# Prevalence of Non-B HIV-1 Subtypes in North Italy and Analysis of Transmission Clusters Based on Sequence Data Analysis

**DOI:** 10.3390/microorganisms8010036

**Published:** 2019-12-23

**Authors:** Giovanni Lorenzin, Franco Gargiulo, Arnaldo Caruso, Francesca Caccuri, Emanuele Focà, Anna Celotti, Eugenia Quiros-Roldan, Ilaria Izzo, Francesco Castelli, Maria A. De Francesco

**Affiliations:** 1Institute of Microbiology, Department of Molecular and Translational Medicine, University of Brescia-Spedali Civili, 25123 Brescia, Italy; giovanni.lorenzin@hotmail.it (G.L.); franco.gargiulo@asst-spedalicivili.it (F.G.); arnaldo.caruso@unibs.it (A.C.); francesca.caccuri@unibs.it (F.C.); 2Institute of Microbiology and Virology, Department of Biomedical, Surgical and Dental Sciences, University of Milan, 20122 Milan, Italy; 3Department of Infectious and Tropical Diseases, University of Brescia and ASST Spedali Civili Hospital, 25123 Brescia, Italy; emanuele.foca@unibs.it (E.F.); anna.celotti@gmail.com (A.C.); maria.quirosroldan@unibs.it (E.Q.-R.); izzo.ilaria@hotmail.it (I.I.); francesco.castelli@unibs.it (F.C.)

**Keywords:** clusters, HIV, phylogeny, subtypes, epidemic

## Abstract

HIV-1 diversity is increasing in European countries due to immigration flows, as well as travels and human mobility, leading to the circulation of both new viral subtypes and new recombinant forms, with important implications for public health. We analyzed 710 HIV-1 sequences comprising protease and reverse-transcriptase (PR/RT) coding regions, sampled from 2011 to 2017, from naive patients in Spedali Civili Hospital, Brescia, Italy. Subtyping was performed by using a combination of different tools; the phylogenetic analysis with a structured coalescence model and Makarov Chain Monte Carlo was used on the datasets, to determine clusters and evolution. We detected 304 (43%) patients infected with HIV-1 non-B variants, of which only 293 sequences were available, with four pure subtypes and five recombinant forms; subtype F1 (17%) and CRF02_AG (51.1%) were most common. Twenty-five transmission clusters were identified, three of which included >10 patients, belonging to subtype CRF02_AG and subtype F. Most cases of alleged transmission were between heterosexual couples. Probably due to strong migratory flows, we have identified different subtypes with particular patterns of recombination or, as in the case of the subtype G (18/293, 6.1%), to a complete lack of relationship between the sequenced strains, revealing that they are all singletons. Continued HIV molecular surveillance is most important to analyze the dynamics of the boost of transmission clusters in order to implement public health interventions aimed at controlling the HIV epidemic.

## 1. Introduction

Recently, UNAIDS/WHO [1] has estimated that 37.9 million of people are infected with HIV worldwide, with about 1.7 million of new infections in 2018. HIV is characterized by a genetic diversity due to the high mutation and recombination rates induced by the HIV RT enzyme and to its high rates of viral replication [2,3]. There are, in fact, four phylogenetic groups of HIV-1 (M, N, O, and P), even if only the M group is responsible for the AIDS pandemic. This group is further divided into subtypes (A, B, C, D, F, G, H, J, and K) and in sub-subtypes (A1, A2, A3, A4, F1, and F2). Recombination of two or more subtypes [4] that co-circulate among high-risk groups in the same geographical area gives rise to unique recombinant forms (URFs) and circulating recombinant forms (CRFs). The number of CRFs is increasing; to date, 98 CRFs have been characterized. The distribution of subtypes and CRFs varies globally. Subtype C is mainly found in Sub-Saharan Africa and India and accounts for 46.6% of HIV infections, followed by subtype B (12.1%), mostly present in Western Europe, the United States, and Australia, and by subtype A (10.3%), which is present in Eastern Europe and Central Asia [5]. CRF02_AG is the fourth prevalent subtype (7.7%), even if it caused about 50% of HIV infections in West and Central Africa [6], followed by CRF01_AE (5.3%), subtype G (4.6%) and D (2.7%) [5].

According to this recent systematic review [5], on a global scale, there was an increase in the proportion of subtype B, and a decrease in the proportion of subtypes C, G, and CRF02_AG, while subtypes A and D were stable. Furthermore, CRF01_AE and other CRFs increased, leading to an increased number of recombinants over time.

This genetic diversity may affect the clinical course because it has been suggested that HIV-1 subtypes may differ in disease progression [7,8,9], in differential therapy outcomes [10,11,12,13], in immune responses to HIV-1 [14,15] and in specific transmission routes [16,17].

However, the HIV-1 subtype geographic distribution is changing, with a consistent increase of non-B infections in regions where subtype B has been prevalent for a long time, such as in Western and Central Europe. This phenomenon is linked to migratory waves that imported them from other geographic locations.

Also, in Italy, a B subtype prevalent area, some different geographically restricted non-B subtypes and CRFs are now penetrating, favored by immigration and travels [18,19,20,21]. A recent study has shown, in fact, an increase of non-B variants from 15.8% to 29.7% during the study period [22].

Phylogenetic analyses in conjunction with traditional epidemiological monitoring can be used to identify HIV transmission clusters and improve our understanding of the dynamics of viral spread. Such tools can be conducted by using large sequence datasets that are routinely collected for clinical care and allow monitoring and characterization of subtype-B and non-B subtypes.

So far, the aim of this paper is to characterize the HIV-1 subtypes diversity in Brescia, North Italy, a geographical area with a high immigration rate from more than 60 different countries (foreigners represented 14.4% of the total population in the Province of Brescia in 2017 [23], an almost double percentage compared to the national one (8.3% in 2016), in order to evaluate trends in non-B subtype prevalence and identify transmission clusters involving non-B subtypes.

## 2. Materials and Methods

### 2.1. Study Population

During the study period (2011–2017), plasma samples from 710 patients newly diagnosed with HIV-1, naive to antiretroviral therapy (ART), attending at Spedali Civili General Hospital of Brescia, in North Italy, were analyzed for antiretroviral drug resistance genotyping, as part of routine clinical practice. The recent infections were estimated by: (a) clinical/laboratory evidence of primary HIV infection such as HIV-1 RNA levels >10000 copies/mL and negative or indeterminate HIV-1 antibody test; (b) a documented negative HIV-1 test preformed within 6 months before the HIV-1 diagnosis; and (c) an antibody avidity index ≤0.8 [24].

Demographic data included gender, age at diagnosis, country of birth, and HIV transmission risk factors; clinical data included CD4+ T-cell count; virologic data included plasma viral load and eventual co-infections. Transmission risk categories were men having sex with men (MSM), heterosexuals, intravenous drug users (IDU), and others (unknown risk factors). All of this information was collected during clinical practice and recorded in an electronic database.

The study was conducted in accordance with the 1964 Declaration of Helsinki and the ethical standards of the Italian Ministry of Health. In detail, as this study was retrospective and non-pharmacological, informed consent was not provided, since, in Italy, ethical authorization for these studies is not required (Italian Guidelines for classification and conduction of observational studies, established by the Italian Drug Agency, “Agenzia Italiana del Farmaco—AIFA” on 20 March 2008). All data were fully anonymized before the statistical analysis was performed.

### 2.2. HIV-1 Subtyping and Sequencing

A contiguous *pol* sequence was generated by spanning the protease and reverse transcriptase regions. Sequences were amplified by using in-house techniques or the TruGene HIV-1 genotyping kit (TruGene Siemens Healthcare Diagnostics GmbH, Eschborn, Germany), according to the manufacturer’s instructions. For the in-house protocol, viral RNA was extracted from 140 μL of the plasma sample, using a QIAamp Viral RNA Mini kit (Qiagen, Milano, Italy) and was suspended in 60 μL of elution buffer. RT-PCR was used to generate a template with the primer set UNI-KS-1 (1817–1844, 10 pmol/mL, forward) and UNI-KS-T/C-2 (3582–3555, 10 pmol/mL, reverse), using a SuperScript III One-Step RT-PCR kit and Platinum Taq DNA polymerase (Invitrogen, Milano, Italy). After RT-PCR, nested PCR was performed. The PCR product was subjected to direct sequencing using an ABI Prism Bigdye Terminator Cycle Sequencing 1.1 Ready Reaction Kit (Applied Biosystems, Milano, Italy) with an automated sequencer (ABI Prism 310 DNA Genetic Analyzer; Applied Biosystems) [25]. The sequenced portion corresponds to the pol region, from position 2253 to 3554 (with some variability linked to the different primers used). Reference genome was HXB2 HIV-1 (GenBank accession number K03455).

Sequences were aligned by using MAFFT v. 7.407 [26] and edited manually by using Bioedit software (v. 7.0.5.3).

Subtyping was performed by using a combination of the following tools; Rega v3, http://regatools.med.kuleuven.be/typing/v3/typingtool (Rega Institute for Medical Research, Leuven, Belgium) [27]; Comet, http://comet.retrovirology.lu/ (Laboratory of Retrovirology, Luxembourg Institute of Health, Luxembourg) [28]; SCUEAL, http://www.datamonkey.org/dataupload_scueal.php (University of California San Diego, La Jolla, CA, US) [29]; jpHMM, http://jphmm.gobics.de/submission_hiv (Institute of Microbiology and Genetics, University of Gottingen, Niedersachsen Germany) [30] and NCBI, http://www.ncbi.nlm.nih.gov/projects/genotyping/formpage.cgi (National Center for Biotechnology Information, Bethesda; city, MD, county) [31]. The final subtype was attributed only in case of concordance of at least 3 tools.

In case of discordance between the various systems, we relied on manual molecular phylogenetic analysis (Mphy).

For Mphy analysis different datasets were used (Subtype reference, 2010, 170 seq; RIP custom background, 2017, 140 seq; Filtered web alignments, 2017, 3099 seq; Custom web alignments, 2017, 4819 seq), in order to obtain sequences belonging to all HIV-1 groups, subtypes, sub-subtypes and CRFs. Where possible, no less than 10 sequences per subtype were integrated into the analysis dataset.

For each subtyping analysis, we generated a maximum likelihood tree (ML), using the IQ-TREE software (1.6.11) in association with ModelFinder. The support for the inferred relationships was evaluated by applying a bootstrap analysis with 1000 replicates. The model generally associated with the analysis was GTR + G (4) + I. When a sequence clustered monophyletically with a bootstrap value >70%, the subtype was assigned.

### 2.3. Determination of Resistance of HIV-1 Subtypes and CRFs

Drug resistance mutations (DRM) were determined by using the latest available version of the HIVDB (v.8.9-1) and Sierra (v.2.4.2) software [32], implemented in the online tool provided by Stanford University (https://hivdb.stanford.edu/hivdb/by-sequences/), by analyzing the consensus obtained from the sequencing of the GAG-POL region of viral isolates from plasmas of HIV-1 positive patients. Major DRMs were selected by using the most recent International AIDS Society (IAS) mutation list [33] and the latest Stanford HIV Drug Resistance Database [34].

### 2.4. Phylogenetic Analyses and Transmission Clusters

All the non-B sequences were further analyzed by checking the recombination breakpoint, using RIP 3.0 software. In order to determine the phylogenetic signal present in the various datasets corresponding to the various CRFs, a likelihood-mapping analysis was performed by using TREE-PUZZLE v5.334 (10,000 randomly chosen quartets) [35]. A likelihood map is formed by an equilateral triangle: each dot within the triangle constitutes the likelihoods of the three possible unrooted trees of four sequences (quartets), randomly selected from the dataset. The dots close to the corners or at the sides respectively represent tree-like (fully resolved phylogenies in which one tree is clearly better than the others) or network-like phylogenetic signals (three regions for which there is no possibility to choose between two topologies). The central area of the map represents a star-like signal (the region where the star tree appears is the optimal tree).

The HIV- BLAST software was used (https://www.hiv.lanl.gov/content/sequence/BASIC_BLAST/basic_blast.html) with the purpose of finding, in the Los Alamos HIV database, the sequences of subtypes and CRFs more similar to those present in our dataset and of integrating them in the analysis to correctly identify monophyletic clusters. For each sequence analyzed in our study, the 10 sequences with greater identity were integrated into the first phylogenetic analysis. All duplicate sequences were previously excluded *p* from the analysis.

We then inferred the phylogeny by using a ML approach, using the ModelFinder, IQ-TREE with 1000 bootstrap analysis replicates (GTR + G + I model in all datasets as inferred from the automated software).

Putative transmission clusters were identified by using the automated tool HIV-TRACE [36] and compared with Cluster Picker v1.2.3, to minimize the bias in transmission cluster detection. We defined clusters as clades with high bootstrap branch support (cut off 95%) and a maximum pairwise genetic distance between all sequences with a < 3.5%. cut off value A threshold was inferred from every final dataset analyzed (e.g., D = 0.015, ranging from 0.01–0.02). Both software has adequately identified the same potential transmission clusters.

When possible, sequences of the appropriate length belonging to transmission events previously identified in the literature were included in the phylogenetic analysis to verify the correct Bayesian inference [37,38].

Using the ML tree with an appropriate root as an input, we analyzed the correlation between genetic distance and isolation date, using TempEst v1.5 software (available from http://tree.bio.ed.ac.uk/). We performed linear regression analyses between the ‘root-to-tip divergence’ and ‘sampling date’ parameters, in order to identify if there was a temporal signal in the different datasets and to infer the best clock signal which maximizes the coefficient of correlation (R).

The datasets which were included in the aforementioned flower selection criteria were subjected to analysis with a Bayesian phylogenetic approach for an estimation of times of common ancestry. The analysis was carried out to highlight the demographic trend of the viral strains and to estimate the date of the possible transmission events in the clusters.

Since the substitution rate and the recombination breakpoints were identical in most of the cases analyzed per single transmission cluster, we used a strict molecular clock as it best fitted our data.

The analysis was conducted by using the software suite included in the Beast2 package (v2.6.0, available from https://www.beast2.org/2019/08/08/what-is-new-in-v2.6.1.html).

In Beauty, the parameters selected were: substitution model GTR + G (4cat) + I, with the “Tip dates” option, under the assumption of a strict molecular clock (evolutionary rate = 1.5 × 10^−3^ substitutions/site/year) and a Bayesian skyline coalescent tree prior. All the other options were calibrated according to the analyzed dataset. At least three independent runs of BEAST2 were performed for each transmission cluster under different demographic models and clock models (strict vs. relaxed), with a MCMC chain length ranging between 8.0 × 10^7^ and 2.0 × 10^8^ states depending on the different datasets analyzed [39].

Markov chain Monte Carlo (MCMC) sampling was run for 80 million generations, with 10% burn-in and sampling every 10,000 generations. The Tracer software (v1.7) [40] was used to visualize and diagnose the MCMC analysis output; MCMC convergence and effective sample sizes (ESS) were taken into consideration if the ESS for each parameter was at least >200.

Maximum Clade Credibility trees were then generated by using TreeAnnotator v1.8.1 and visualized with FigTree v1.4.4 (http://tree.bio.ed.ac.uk/software/figtree/) or iTOL-v4 (https://itol.embl.de/).

### 2.5. Statistical Analysis

The statistical analysis was firstly performed by grouping the variables into different groups using a hierarchical clustering approach, and then a total intra-cluster variance with a k-means algorithm was minimized.

Univariate analyses were performed, using Pearson’s Chi-square for categorical variables and the nonparametric Mann–Whitney U test for continuous variables. The variables with a *p*-value ≤ 0.05 in the univariate analyses were included in the multivariate analysis (ANOVA and Multinomial Logistic Regression/Hosmer–Lemeshow goodness-of-fit test).

The trend of the prevalence of non-B subtypes over time was evaluated by using the chi-square test for the trend (Armitage test) and a multinomial logistic regression model based on year, sex, and sampling region as explanatory variables.

All statistical analyses were performed by using the IBM-SPSS Statistics software (v.25.0.0.1, 64 bit) and the Rstudio (v.3.5.1).

### 2.6. Sequence Data Availability

Because submission of the entire sequence data set to public databases would permit transmission networks to be identified and thus risk breaching patient confidentiality, following scientific and ethical reasons explained in other studies [41,42,43], we have submitted a random sample of 10% (representative of each subtype) to GenBank under accession numbers MN833139–MN833168. However, all other sequences are available on request.

## 3. Results

### 3.1. Study Population

Between 2011 and 2017, 710 subjects were newly diagnosed, and plasma was obtained within one month from the first contact after HIV-1 diagnosis. Of these patients, 304 (43%) had a non-B HIV-1 subtype, of which only 293 sequences were available. Baseline demographic characteristics of the study population are summarized in Table 1. Men were 64.8% of the study population, even if women were represented more in A1 and CRF09_cpx HIV 1 subtypes. The mean age was 39.6 years. Of the 293 patients, 157 (53.5%) were from Italy, 99 (33.7%) were from Africa, 16 (5.4%) were from Eastern Europe, 16 (5.4%) were from Asia, 4 (1.3%) were from South America, and 1 (0.3%) was from North America. Lower CD4 count and higher viral loads were observed in F1 and CRF12_BF HIV-1 subtypes. The most common transmission route was heterosexual contact (205/293, 70%), followed by MSM (53/293, 18%) and IDU (9/293, 3%), while 26 (8.8%) were of unknown transmission route. Trends showed that the proportion of non-B subtypes among all sequences (B and non-B subtypes) increased over time, even if not statistically significant by 2013. In fact, the non-B subtype prevalence, based on the first available sequence, ranged, in fact, from 38% in 2013 to 52% in 2016 and 2017 (*p* = 0.19 for chi-squared trend)

### 3.2. HIV-1 Genotyping

Subtyping tools reported a concordant subtype for 283 (96.5%) sequences. The ten sequences with different subtype assignments all involved recombinants (CRF12_BF and CRF09_cpx).

Subtype distribution was CRF02_AG (150, 51.2%), F1 (50, 17%), C (27, 9.2%), G (18, 6.1%), CRF01_AE (17, 5.8%), CRF06_cpx (12, 4%), CRF12_BF (5, 1.7%), and CRF09_cpx (5, 1.7%).

The phylogenetic tree constructed with reference alignments and all non-B study sequences is depicted in Figure 1.

### 3.3. Transmission Cluster Analysis

Phylogenetic analysis identified 25 putative clusters, with sizes ranging from 2 to 47 patients (Table 2). In total, 147/293 (50.1%) patients were included in transmission clusters, with 84 patients (20%) included in large clusters of 10 or more individuals and belonging to CRF02_AG and F1 subtype, probably because these two subtypes included the larger number of patients. Most clusters (13/25) were pairs; nine clusters included 3–8 individuals; and three large clusters included 10, 27, and 47 individuals (Table 2).

Only one cluster with three individuals included LANL reference. In the Beast analysis, all 25 clusters were highly supported by using a cut-off genetic distance of 0.015, with a bootstrap support of >98% (MCMC posterior probability density is reported in the figures as colored branches; the probability was visualized and interpreted by using DensiTree and Tracer). The comparison between the clustered and non-clustered patients is shown in Table 3. According to the multivariate analysis, the factors associated with being part of a cluster (regardless of size) were Italian in origin (*p* = 0.002) and being infected with a subtype CRF02_AG or F virus (*p* = 0.002). Male gender, transmission risk, higher viral load, and CD4 counts were not identified as independent predictors in multivariate analysis. Factors associated with non-clustered patients were African or Asian origin, a subtype G or A1 virus infection and having more transmitted drug-resistance mutations.

Non-clustered patients belonging to G subtype were mostly from different countries of Africa (16/18, 89%), with just two individuals from Italy, who were interspersed in this group and were heterosexuals; meanwhile, non-clustered patients belonging to A1 subtype included three males and six females. They were heterosexuals and from Italy (4/9, 44.4%), Africa (2/9, 22.2%), Eastern Europe (2/9, 22.2%), and Asia (1/9, 11.1%).

The most frequent nucleoside reverse-transcriptase inhibitor (NRTI) resistance mutations were V106I (50%), M41L (13.3%), and T215D/S (6.6%). The most frequent non-nucleoside reverse-transcriptase inhibitor (NNRTI) mutations were K103N (16.6%) and K101Q/R (10%). The most frequent protease inhibitor (PI) mutations were V82I (66.6%), M46L (10%), and I54V (6.6%).

### 3.4. CRF02_AG Transmission Clusters

CRF02_AG subtype comprises 150 individuals, and the phylogenetic analysis detected four transmission pairs, three transmission clusters which included three or four individuals, and two large transmission clusters of 47 and 27 individuals, respectively. Most of the patients of the four transmission pairs and of the others with three to four individuals were male/female (6/7), while one cluster included only men.

The large cluster of 47 patients comprised heterosexual individuals (91.4%) from Italy (Figure 2b), who are linked with a high, well-supported degree of connectedness 98% (bootstrap procedure standard error (SE): 2%), while the other large cluster of 27 patients comprised mostly Italian individuals (89%), where the dominant transmission route was MSM (70.3%) (Figure 3b). The likelihood mapping analysis of the HIV1 CRF02_AG dataset of these two large clusters showed that the percentage of dots falling in the central area of the triangles was 2.9% and 4.1% (Figure 2a and Figure 3a), thus indicating a > 90% fully resolved phylogenetic signal and a tree-like structure with correct phylogenetic signal.

To describe the temporal patterns of CRF02_AG regional sub-epidemics in Brescia, we carried out phylodynamic analysis. There was evidence for temporal signal in the sequences found within the two large CRF02_AG, as tested by the TempEst program (R^2^ = 0.95). Molecular clock analysis suggested that the time to most recent common ancestor (tMRCA) of the CRF02_AG subepidemic was in 2000 (median estimate; 95% Highest Posterior Density-HPD interval 1992–2010) for the heterosexual individuals (Figure 2c) and in 1997 (median estimate; 95% Highest Posterior Density–HPD interval: 1990–2003) for the MSM group (Figure 3c). The tMRCA should be considered the approximate time of infection of the potential founder of these two CRF02_AG clusters in Brescia sampled in our data.

The mean tMRCA of these clusters range from 5 to 11 years, respectively. All non-clustered patients belonging to CRF02_AG subtype (Figure 4) were mostly from different countries of Africa (50/58, 86.2%) and had heterosexual contact (49/58, 84.4%) as main risk of infection transmission. African patients were mostly from Ghana (17/50, 34%), Nigeria (9/50, 18%), Ivory Coast (9/50, 18%), Cameroon (3/50, 6%) Burkina Faso (3/50, 6%), Senegal (2/50, 4%), Togo (2/50, 4%), Gambia (1/50, 2%), Liberia (1/50, 2%), Sudan (1/50, 2%), Guinea (1/50, 2%), and Morocco (1/50, 2%).

### 3.5. F1 Transmission Clusters

The F1 subtype comprises 50 individuals. Four transmission pairs were identified, two medium clusters involved eight and seven individuals, and one large cluster included 10 individuals (Figure 5). Of the four transmission pairs, seven individuals were from Italy, and one individual was from Africa. Of these, only one was constituted by male/female, while three only involved men. Regarding the two medium and the one large clusters, most of the individuals (23/25, 92%) were from Italy, and the dominant transmission route was through heterosexual contact (21/25, 84%).

Non-clustered patients came from Italy (13/17, 76.4%), America (2/17, 11.7%), Africa (1/17, 5.8%), and Asia (1/17, 5.8%). They comprised 14 males and three females, with mostly MSM (53%).

### 3.6. C Transmission Clusters

The C subtype comprises 27 individuals. Phylogenetic analysis identified five transmission pairs. Of these, four were male/female and one included only men (Figure 6). Three transmission pairs were from Italy and one from Asia. Non-clustered patients were from Italy (9/17, 53%), Asia (6/17, 35%), and Africa (2/17, 12%). They were 11 males and six females, and they were mostly heterosexuals. (10/17, 58.8%).

### 3.7. CRF01_AE Transmission Clusters

CRF01_AE subtype comprises 17 individuals and two transmission clusters were identified. Of these, one included three females and one male, and they were all from Eastern Europe and heterosexuals; the other one included one male from Asia and MSM derived from our dataset, and the other two males derived from LANL reference that were also from the same part of Asia (China) and MSMs, too, highlighting that the country of infection was probably China and this virus was later imported to Italy (Figure 7).

Non-clustered patients were from Italy (4/12, 33.3%), Asia (3/12, 25%), Africa (3/12, 25%), and Eastern Europe (2/12, 16.6%). They included seven males and five females, and they were mostly heterosexuals (8/12, 67%).

### 3.8. CRF06_cpx and CRF12_BF Transmission Clusters

CRF06_cpx subtype comprises 12 individuals, and one transmission cluster was identified. This cluster included four individuals. They were a male/female from Africa, one male from Italy, and one female from America. Non-clustered patients included four males/females and were from Africa (5/8, 62.5%) and from Italy (3/8, 37.5%).

CRF12_BF subtype comprises five individuals, and one transmission cluster was identified. This cluster included 3 individuals. They were two males from Italy and one female from Eastern Europe. Non-clustered patients involved two men, one from Italy and one from Eastern Europe.

## 4. Discussion

We studied the genetic diversity of 710 HIV-1 *pol* sequences obtained from 2011 and 2017 from naive patients by using a combination of phylogenetic and demographic analyses. We found that the overall prevalence of non-B subtypes was 43%, with an increase, even if not statistically significant, from 38% in 2013 to 58% in 2017.

Different Western European countries, in particular those with large Sub-Saharan African immigrant communities, have reported an increase in HIV-1 non-B infections, which are mostly acquired by heterosexual contacts. In fact, higher prevalence of non-B subtypes over time have been documented over time in Belgium [44], Spain [45], France [46], Sweden [47], Luxembourg [48], and Italy [49]. We observed a high degree of viral diversity between non-B subtypes, with six pure subtypes comprising F1, G, C, and A1 (104/293, 35%), which highlights the multiple viral introduction in the region. Two pure subtypes (D and K) were sporadically present as a single strain each (0.03%), and they have not been included in the analysis.

We also found a higher heterogeneity among the CRFs, including CRF02_AG, CRF06_cpx, CRF01_AE, CRF_12BF, and CRF09_cpx (189/293, 65%).

We found that more than half of non-B subtypes (157/293, 53.5%) of our study population included ART-naive patients born in Italy, indicating that non-B strains have become endemic among the Italian population; this finding agreed with the results obtained by a previous work of ours [50] and a recent study [22]. However, travel anamnesis has not been investigated; therefore, it is impossible to know if these patients received the infection in Italy or overseas.

Moreover, we found evidence for several local transmission clusters, including non-B subtypes, of sizes ranging from 2 to 47 individuals. Even if heterosexual contact represented the most (70%) of the infections, MSM transmission accounts for a significant proportion of patients (18%), while the intravenous drug users is poorly represented as risk factor in subjects harboring non-B subtypes population (3%). The finding that 68% of transmission clusters have a small size (*n* = 2 individuals) indicates that they might originate from importation (with a part of the transmission chain which resides in country of origin) or that a part of these clusters has not been diagnosed.

Local transmission was important for Italian natives and infected with CRF_02AG or F1 subtypes, who were more likely to be part of a transmission cluster than other risk groups.

The finding that persons belonging to a transmission cluster were less likely to have major DRM suggests that a higher proportion could have more recent HIV diagnoses compared to patients who were not part of a cluster.

While a great diversity among non-B subtypes was identified in our study, 31.3% of individuals in non-B transmission clusters were infected with CRF02_AG strains.

The prevalence of CRF_02 AG has been estimated to be 3% in Western and Central Europe and North America, representing the most frequent CRF [5] Different proportions of monophyletic clusters involving CRF02_AG have been reported: 29.8%, 27.3%, 25.4%, and 18.2% in Germany, Norway, Austria, and Italy, respectively [51]. These findings underline a common way of CRF02_AG dispersal in Europe, where, even if this subtype has been associated with highly endemic areas [52] representing their putative source, a considerable proportion of their sequences fall within monophyletic clusters.

In our study, we found two large transmission clusters belonging to CRF02_AG subtype: one with 47 heterosexual Italian individuals and another one with 27 MSM Italian individuals, providing evidence for the spread of non-B subtypes among the well-known high-risk population. The transmission chains identified belong to different individuals linked by different risk factors. Given the reduced genetic distance between the analyzed isolates (D < 0.015), the confidence level (bootstrap value > 98%) and the risk factors taken into consideration in relation to geography, the evolutionary model applied in the analyses shows how transmission events can be correlated between them. The temporal signal is well represented, with a low divergence and an excellent linear correlation (R^2^ = 0.95). In the analysis of large clusters, however, this is not sufficient to exclude that there may be subjects which are not sampled and not present within the transmission chain. The analysis provided has the purpose to represent the evolution within a cluster of people with a strong association of predisposing and geographical factors and therefore determine relationship into the sources of an epidemic of ongoing viral spread, at the level of the population. Another large cluster was identified in F1 subtype, with 10 heterosexual Italian individuals, a finding which had already been reported for this subtype [18].

Subtype C comprised small transmission clusters (five pairs), who were heterosexual Italian individuals.

Small transmission clusters were identified for CRF01_AE, CRF06_cpx, and CRF12_BF. CRF01_AE comprised two transmission clusters: one with four individuals originating from Eastern Europe and one with three individuals from Asia, while CRF06_cpx and CRF12_BF comprised foreign and Italian individuals, reflecting all immigration patterns.

G and CRF09_cpx subtypes were associated with non-clustered patients who were mostly from Africa (23/32, 72%) and had heterosexual contact as the principal risk factor to acquire HIV-1 infection, indicating that these subtypes were not spreading in local transmission.

Regarding the A1 sequences isolated in Brescia, we found that most of them (44.4%) were from Italy and only a minority was found in foreigners from Africa (22.2%) and Eastern Europe (22.2%) that not clustered, underlining that these cases are likely to be infections acquired abroad by heterosexual contact and with limited diffusion in Brescia.

Regarding CRF09_cpx subtypes found in this study, they have a mosaic structure due to multiple recombination events with different parental subtypes. Different new CRFs have been described in Europe, as various non-B subtypes circulate together with the founder B clade and are identified with increasing prevalence in different countries [53].

So far, continued HIV-1 molecular surveillance will be essential for understanding the dynamics of expansion of transmission clusters. This comprehension might allow us to focus preventive efforts in populations with the highest risk of HIV-1 acquisition and continuing transmission [54,55,56,57,58]. Furthermore, it is important in order to monitor the efficacy of public health measure aimed at controlling the epidemic [59]. The identification of clusters that contain transmission events and their correlation with epidemiological data such as geographic, behavioral, and biological factors will help to develop targeted interventions in infection prevention and control. The ultimate goal in identifying transmission clusters is the possibility to interrupt these clusters with targeted interventions.

## Figures and Tables

**Figure 1 microorganisms-08-00036-f001:**
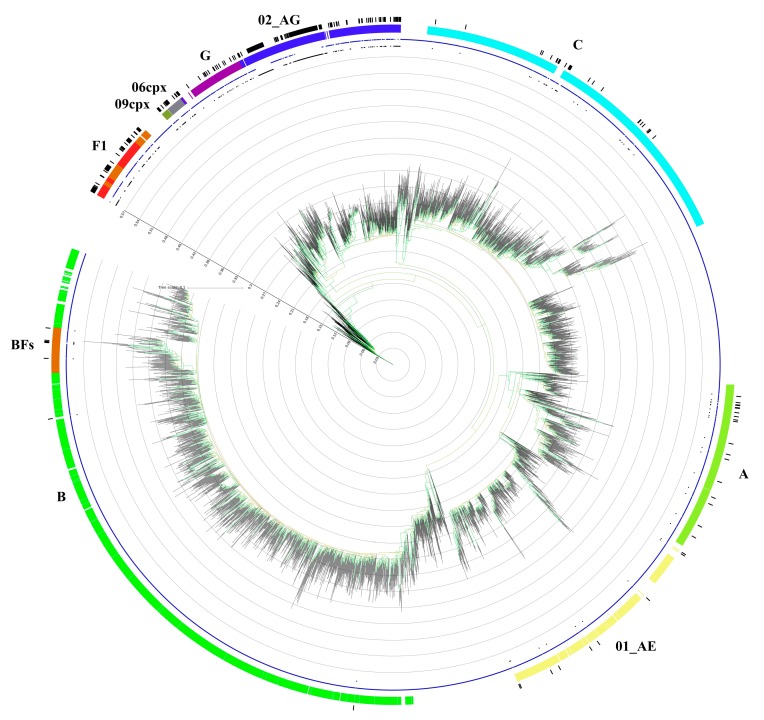
Phylogenetic tree generated by using the GTR + G + I subsidence model and 1000 replicated bootstrap analysis with the IQ-Tree software. The dataset (which includes 293 sequences originating from antiretroviral therapy-naive patients) is compared to a custom sequence database composed of all the reference sequences present in the LosAlamos database (3497 total aligned sequences from reference sequences and filtered web alignments). The colored ring represents only the subtypes found in the study population, and the black ring represents the samples taken into consideration. The blue star symbols represent the reference sequences, and the black circle symbols represent the analyzed samples. The bootstrap value is represented by the color of the branches; the minimum value is 60, and it is represented in red by scaling the gradient up to 100, which is colored green.

**Figure 2 microorganisms-08-00036-f002:**
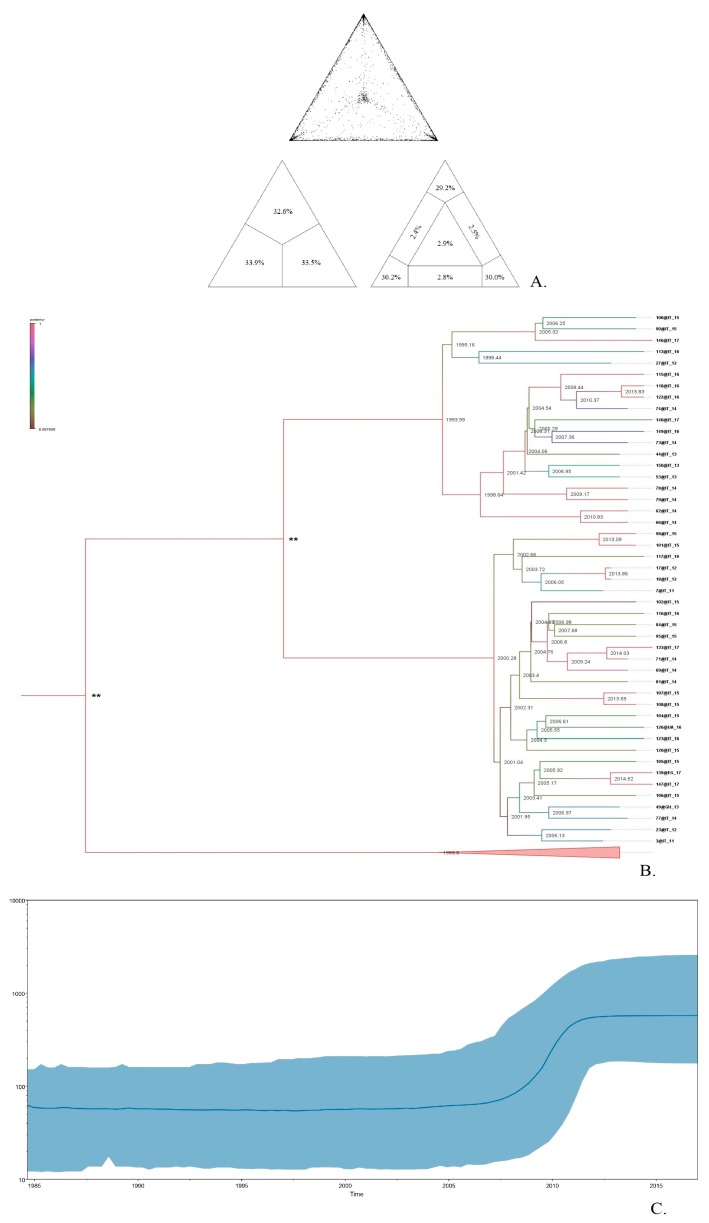
Cluster transmission of CRF02_AG, including Italian heterosexual individuals. The upper part (**A**) shows the Likelihood mapping of the analyzed sequences used for the phylogenetic tree reconstruction, based on 10,000 random quartets. The central triangle is the probability of obtaining a star-type topology. The central part (**B**) represents the consensus tree obtained visualized with FigTree, and the colored branches represent the posteriority, to each node the corresponding node age is associated. The lower part (**C**) represents historical demographic trends in sampled with a Bayesian skyline plot; on the x axis, the date for sampling year is shown; on the *y* axis the effective population size is shown.

**Figure 3 microorganisms-08-00036-f003:**
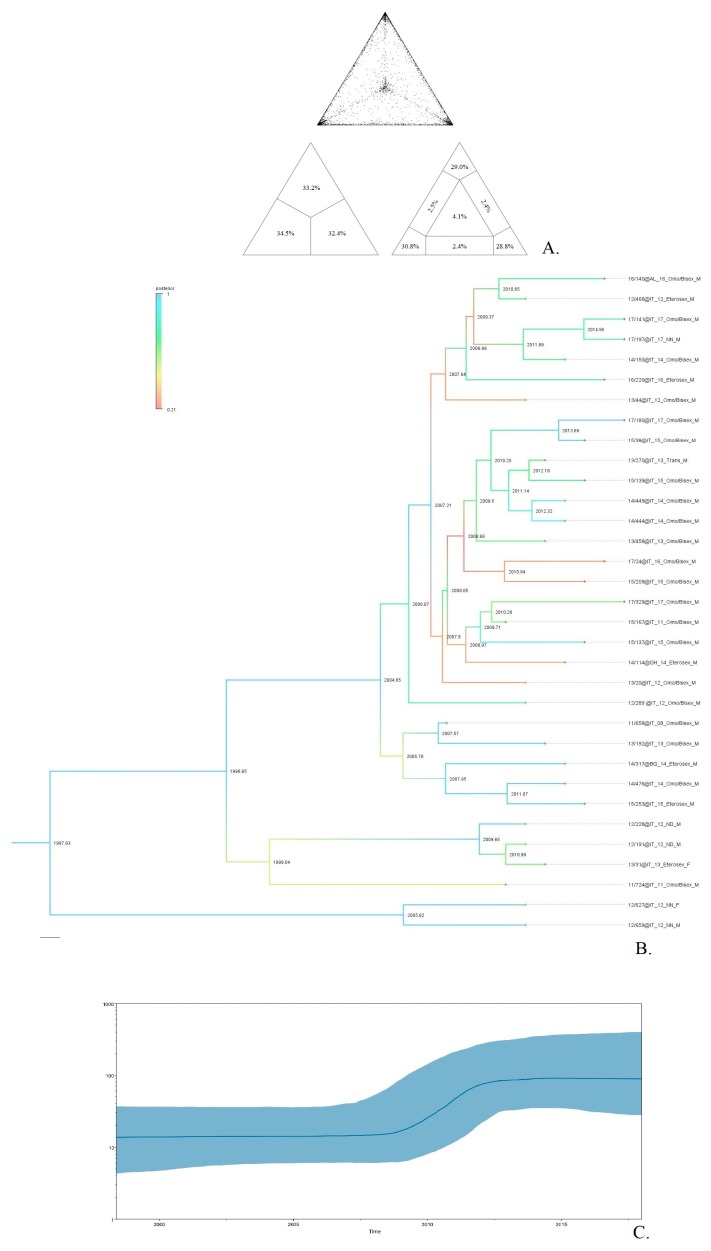
Cluster transmission of CRF02_AG, including Italian MSM. The upper part (**A**) shows the likelihood mapping of the analyzed sequences used for the phylogenetic tree reconstruction, based on 10,000 random quartets. The central triangle is the probability of obtaining a star-type topology. The central part (**B**) represents the consensus tree obtained visualized with FigTree; the colored branches represent the posteriority, and to each node, the corresponding node age is associated. The lower part (**C**) represents historical demographic trends in sampled, with a Bayesian skyline plot; on the x axis, the date for sampling year is shown; on the *y* axis the effective population size is shown.

**Figure 4 microorganisms-08-00036-f004:**
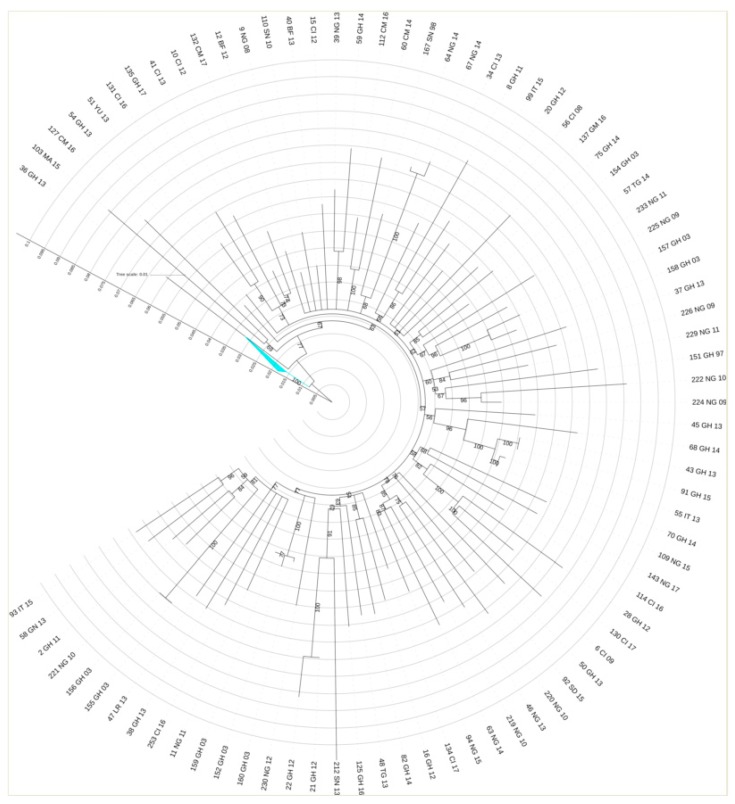
Maximum likelihood tree obtained from the CRF02_AG sequences that did not present transmission events related to the other two transmission clusters. The substitution model is GTR + G + I; bootstrap analysis with 1000 replicates was applied. This tree highlights how the analysis of sequences originating from patients of African origin makes it impossible to identify a unique evolution/transmission model, given the genetic distance between the isolates. The country of origin of sequences is indicated by a two-letter code: GH, Ghana; CM, Cameroon; Ng, Nigeria; SN, Senegal; CI, Ivory Coast; GM, Gambia; MA, Morocco; SD, Sudan; LR, Liberia; Tg, Togo; BF, Burkina Faso, and GN, Guinea.

**Figure 5 microorganisms-08-00036-f005:**
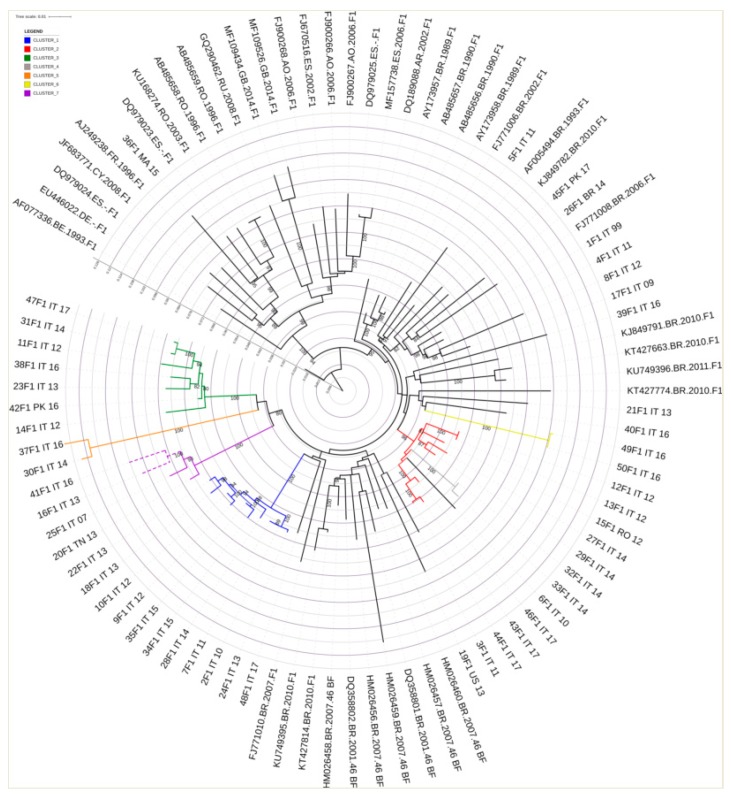
Phylogenetic inference derived from sequence analysis of the subtype F1. Maximum likelihood phylogenetic tree was built by using the iq-tree software and displayed via i-Tol. The substitution model used was GTR + G + I, with 1000 replicates in bootstrap analysis. Branch support values are shown for each tree, consisting of related bootstrap proportions. Compressed clades represent the control group for correct phylogeny inference. Colored branches indicate Italian transmission clusters.

**Figure 6 microorganisms-08-00036-f006:**
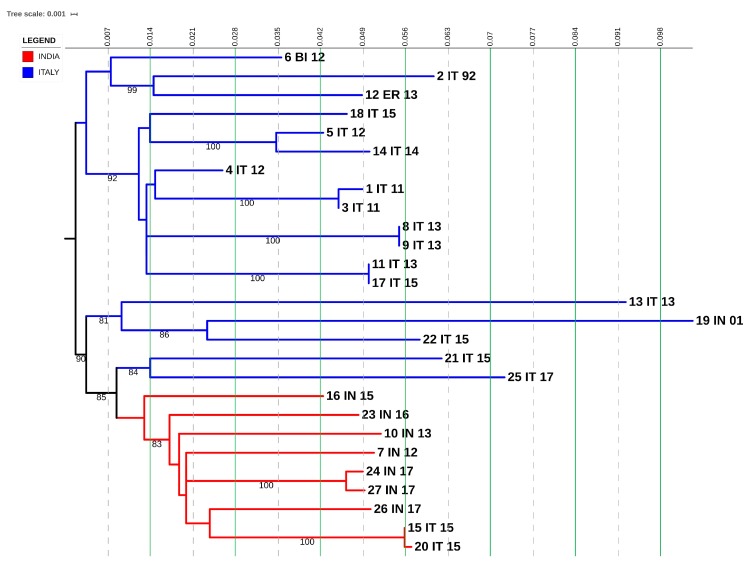
Phylogenetic inference derived from sequence analysis of the subtype C. Maximum likelihood phylogenetic tree built using the iq-tree software and displayed via i-Tol. The substitution model used is GTR + G + I, with 1000 replicates in bootstrap analysis. Branch support values are shown for each tree, consisting of related bootstrap proportions. Compressed clades represent the control group for correct phylogeny inference.

**Figure 7 microorganisms-08-00036-f007:**
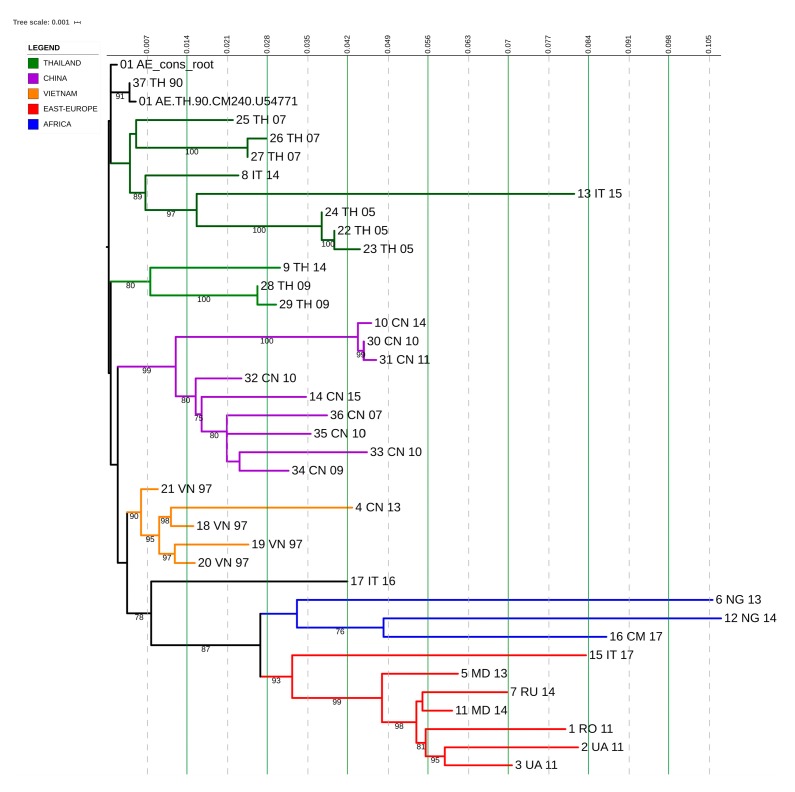
Phylogenetic inference derived from sequence analysis of the subtype CRF01_AE. Maximum likelihood phylogenetic tree was built by using the iq-tree software and displayed via i-Tol. The substitution model used was GTR + G + I, with 1000 replicates in bootstrap analysis. Branch support values are shown for each tree, consisting of related bootstrap proportions. Compressed clades represent the control group for correct phylogeny inference.

**Table 1 microorganisms-08-00036-t001:** Baseline characteristics of HIV-1 infected patients during the study period, according to different HIV-1 non-B subtypes.

Characteristics	HIV-1 Subtypes									
	F1 *n* = 50	C *n* = 27	G *n* = 18	A1 *n* = 9	CRF_02AG *n* = 150	CRF06_CPX *n* = 12	CRF01_AE *n* = 17	CRF12_BF *n* = 5	CRF09_CPX *n* = 5	Total
Gender										
Male	37 (74%)	17 (63%)	9 (50%)	3 (33%)	104 (69%)	6 (50%)	9 (53%)	4 (80%)	1 (20%)	190 (65%)
Female	13 (26%)	10 (37%)	9 (50%)	6 (67%)	46 (31%)	6 (50%)	8 (47%)	1 (20%)	4 (80%)	103 (35%)
Age (years), Median (IQR)	46 (17)	42 (16)	40 (15)	43 (18)	41 (18)	41 (15)	37 (18)	35 (19.5)	32 (17)	
Risk factors, *n* (%)										
MSM	9 (18%)	6 (22%)	1 (6%)	1 (11%)	29 (19%)	2 (17%)	5 (29%)	0	0	53 (18%)
Heterosexual	32 (64%)	15 (56%)	17 (94%)	6 (67%)	108 (72%)	10 (83%)	10 (59%)	3 (60%)	4 (80%)	205 (70%)
IDU	3 (6%)	1 (4%)	0	1 (11%)	3 (2%)	0	0	1 (20%)	0	9 (3%)
Others	6 (12%)	5 (18%)	0	1 (11%)	10 (7%)	0	2 (12%)	1 (20%)	1 (20%)	26 (9%)
CD4 count, mean (SD) cells/mL	253 (257)	396 (310)	325 (300)	453 (311)	330 (233)	403 (280)	260 (147)	228 (261)	227 (193)	
CD4 count < 200 cell/mL, *n* (%)	27 (54%)	7 (26%)	7 (39%)	2 (22%)	48 (32%)	3 (25%)	6 (35%)	3 (60%)	2 (40%)	
PVL, mean copies/mL	733,602	380,374	153,839	366,040	304,435	38,707	151,028	628,462	43,099	
Geographic origin, *n* (%)										
Italy	43 (86%)	17 (63%)	2 (11%)	4 (44%)	80 (53.3%)	4 (34%)	4 (23.5%)	3 (60%)	0	157 (54%)
Europe	1 (2%)	0	0	2 (22%)	5 (3%)	0	6 (35%)	2 (40%)	0	16 (5%)
America	2 (4%)	0	0	0	2 (1%)	1 (8%)	0	0	0	5 (2%)
Africa	2 (4%)	2 (7%)	16 (89%)	2 (22%)	62 (41%)	7 (58%)	3 (18%)	0	5 (100%)	99 (34%)
Asia	2 (4%)	8 (30%)	0	1 (11%)	1 (0.7%)	0	4 (57.1%)	0	0	16 (5%)
Coinfection with HCV	7 (14%)	1 (4%)	0	1 (11%)	9 (6%)	0	1 (6%)	1 (20%)	0	20 (7%)
Coinfection with HBV	2 (4%)	1 (4%)	4 (22%)	1 (11%)	12 (8%)	2 (17%)	1 (6%)	0	1 (20%)	24 (8%)

Abbreviations: IQR; interquartile range; PVL, plasma viral load; MSM, men having sex with men; IDU, intravenous drug users.

**Table 2 microorganisms-08-00036-t002:** Distribution of HIV-1 infected patients according to the different HIV-1 non-B subtypes and their clustering status.

	HIV Subtypes								
	F1 *n* = 50	C *n* = 27	G *n* = 18	A1 *n* = 9	CRF_02AG *n* = 150	CRF06_CPX *n* = 12	CRF01_AE *n* = 17	CRF12_BF *n* = 5	CRF09_CPX *n* = 5
No cluster	17 (34%)	17 (63%)	18 (100%)	9 (100%)	58 (38%)	8 (67%)	12 (70.5%)	2 (40%)	5 (100%)
Small Cluster (2–3)	8 (16%)	10 (37%)	0	0	17 (11.3%)	4 (33%)	4 (23.5%)	3 (60%)	0
Medium Cluster (4–9)	15 (30%)	0	0	0	1 (0.6%)	0	1 (5.8%)	0	0
Large cluster (≥10)	10 (20%)	0	0	0	74 (50%)	0	0	0	0

**Table 3 microorganisms-08-00036-t003:** Univariate and multivariate analysis for comparison between patients on isolated branches in the phylogenetic tree and patients in phylogenetic clusters.

	Univariate Analysis			Multivariate Analysis	
	Not in Cluster *n* = 146	In Cluster *n* = 147	*p*-Value	OR (95% CI)	*p*-Value
Gender					
Male	82 (56%)	108 (73.4%)	**0.0022**	0.73 (0.34–1.58)	0.43
Female	64 (44%)	39 (26.5%)			
Age (years), Median (IQR)	39 (17)	40 (18)	0.45		
Risk factors, *n* (%)					
MSM	24 (16%)	29 (19.7%)	0.54		
Heterosexual	106 (73%)	99 (67.3%)	0.37		
IDU	4 (3%)	5 (3.4%)	1		
Others	12 (8%)	14 (9.5%)	0.83		
Geographic origin, *n* (%)					
Italy	40 (27%)	117 (79.5%)	**0.0001**	**8.73 (1.33–57)**	**0.02**
Europe	7 (5%)	9 (6.1%)	0.79		
America	3 (2%)	2 (1.3%)	0.68		
Africa	84 (58%)	15 (10.2%)	**0.0001**		
Asia	12 (8%)	4 (2.7%)	**0.04**		
Subtypes					
F1	17 (11.6%)	33 (22.4%)	**0.019**	**6.17 (1.24–30.73)**	**0.026**
G	18 (100%)	0	**0.0001**		
C	17 (11.6%)	10 (6.8%)	0.16		
A1	9 (100%)	0	**0.001**		
CRF02_AG	58 (40%)	92 (60%)	**0.0001**	**0.093 (0.02–0.43)**	**0.02**
CRF06_CPX	8 (5.4%)	4 (2.7%)	0.25		
CRF01_AE	12 (8.2%)	5 (3.4%)	0.08		
CRF12_BF	2 (1.3%)	3 (2%)	1		
CRF09_CPX	5 (100%)	0	**0.03**		
CD4 count, mean (SD) cells/mL	332 (244)	354 (252)	0.87		
PVL, mean copies/mL	184,300	519,463	**0.01**	1 (Ref)	0.2
Transmission drug resistance	24 (16.4%)	9 (6.1%)	**0.005**		

Abbreviations: IQR; interquartile range; PVL, plasma viral load; MSM, men having sex with men; IDU, intravenous drug users; OR, odds ratio; CI, confidence interval; Ref, reference category; bold characters indicate statistical significance.

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
