# Peer review of "Prevalence of Non-B HIV-1 Subtypes in North Italy and Analysis of Transmission Clusters Based on Sequence Data Analysis"

_microorganisms, 2019, doi:10.3390/microorganisms8010036_

Round 1

Reviewer 1 Report

Giovanni Lorenzin et colleagues analysed 710 HIV-1 pol (PR/RT) sequences ns, sampled from 2011 to 2017 from newly infected ART-naive patients in Spedali Civili’s Hospital, Brescia, Italy and they  investigated the prevalence of non-B infections in the region and the molecular epidemiology of those non-B infections.

Specific comments and suggestions

Is not clearly stated why 9 non-B sequences were not available for the analysis. All 710 sequences (B and non-B ones) should be submitted in Genbank and authors should provide their relevant accession numbers, if already deposited. Recombination analysis of 5 putative CRF09_cpx strains does not add any important info to this work and does not make sense since only pol sequences (only ¬1000nt according to fig 2) were used for the recombination analysis. Full length or near full length sequences would be far more imformative.   

Author Response

Reviewer 1

Giovanni Lorenzin et colleagues analysed 710 HIV-1 pol (PR/RT) sequences ns, sampled from 2011 to 2017 from newly infected ART-naive patients in Spedali Civili’s Hospital, Brescia, Italy and they  investigated the prevalence of non-B infections in the region and the molecular epidemiology of those non-B infections.

Specific comments and suggestions:

Is not clearly stated why 9 non-B sequences were not available for the analysis

Answer: The quality of these sequences was low and we have decided to exclude them from the analysis

All 710 sequences (B and non-B ones) should be submitted in Genbank and authors should provide their relevant accession numbers, if already deposited

Answer: For the study, we retrieved from the archive only sequences belonging to non B subtypes. Because submission of the entire sequence data set to public databases would permit transmission networks to be identified and thus risk breaching patient confidentiality, following scientific and ethical reasons explained in other studies {Alizon et al (2010), Kouyos et al (2010), Esbjornsson et al. (2016)} we have submitted a random sample of 10% (representative of each subtype) to GenBank (with upcoming) under accession numbers. However, all other sequences are available on request. This sentence has been added in the manuscript in the Materials and Methods as “Sequence data availability” subsection

Recombination analysis of 5 putative CRF09_cpx strains does not add any important info to this work and does not make sense since only pol sequences (only ¬1000nt according to fig 2) were used for the recombination analysis. Full length or near full length sequences would be far more imformative.   

Answer: We agree with the referee’s suggestion and we have deleted this part from Material and Methods and Results section; we have also deleted the corresponding Figure 2.

Reviewer 2 Report

This manuscript presents a study investigating HIV-1 subtypes in North Italy. There are 3 stated aims: (1) to characterise HIV-1 subtype diversity in Brescia; (2) to evaluate trends in non-B subtype prevalence and (3) to identify transmission clusters involving non-B subtypes. The study focuses on non-B subtypes which comprise 43% of the strains overall and records the prevalence of the 9 non-B subtypes identified from study participants.  Further the manuscript relates an investigation of transmission clusters in each of the non-B subtypes and correlates clustering with risk factors and geographic origin.

Overall this study has used appropriate methods throughout for this investigation and it has been performed to a high standard. The following points however require attention.

Major points

The figures are of a poor quality and it is impossible to see details which become blurred on zooming in. In particular I found it very difficult to interpret figure 1. Some of the content is too small, for example the scale bars. The legends for figure 3 and 4 are identical - can the clusters included in each of them be distinguished in some way by including some information in the legend? The figures or legends would also benefit from additional information including indicating the most important demographic information e.g. figure 3, Italian heterosexuals and figure 4, Italian MSM. Some of the tree branches are coloured however there is not a key for these colours. As geographic factors are very important in the conclusions of this study, it would be better if this was somehow depicted in the trees e.g. colour-coded according to country or region where appropriate. More geographic information on the isolates of African origin, if available, should be included in text and figures. Some of the trees with less branches might be better as linear trees as these are easier to read e.g. figure 7 and 8. The English within the text needs significant improving, in particular improvements in grammar and word choice are required. I cannot find mention of sequence submission to a database such as NCBI. This should be done and the GenBank accession numbers included in the text. More details on sequence amplification and sequencing should be included in the methods section.

Minor points

Results of the univariate analysis are shown in table 3 but the multivariate analysis is given in the text. I found this confusing and think it would be better if the multivariate analysis was also included in the table. In table 2, only the subtypes with more samples have large clusters and it is likely that this is a sampling issue. This should be pointed out in the text. In table 1, it would be helpful to have a final column with the total or overall figures.

Author Response

Reviewer 2

This manuscript presents a study investigating HIV-1 subtypes in North Italy. There are 3 stated aims: (1) to characterise HIV-1 subtype diversity in Brescia; (2) to evaluate trends in non-B subtype prevalence and (3) to identify transmission clusters involving non-B subtypes. The study focuses on non-B subtypes which comprise 43% of the strains overall and records the prevalence of the 9 non-B subtypes identified from study participants.  Further the manuscript relates an investigation of transmission clusters in each of the non-B subtypes and correlates clustering with risk factors and geographic origin.

Overall this study has used appropriate methods throughout for this investigation and it has been performed to a high standard. The following points however require attention.

Major points

The figures are of a poor quality and it is impossible to see details which become blurred on zooming in. In particular I found it very difficult to interpret figure 1. Some of the content is too small, for example the scale bars

Answer: We have sent all the figures in high resolution (300dpi) as supplementary material.

The legends for figure 3 and 4 are identical - can the clusters included in each of them be distinguished in some way by including some information in the legend? The figures or legends would also benefit from additional information including indicating the most important demographic information e.g. figure 3, Italian heterosexuals and figure 4, Italian MSM

Answer: We agree with the referee’s suggestion and we differentiate Figure legends including demographic information

More geographic information on the isolates of African origin, if available, should be included in text and figures

Answer: We agree with the referee’s suggestion and we have added information about the geographic origin of African patients both in the manuscript and in the figure legend

Some of the tree branches are coloured however there is not a key for these colours. As geographic factors are very important in the conclusions of this study, it would be better if this was somehow depicted in the trees e.g. colour-coded according to country or region where appropriate.

Answer: We agree with the referee’s suggestion and we have added a figure legend.

Some of the trees with less branches might be better as linear trees as these are easier to read e.g. figure 7 and 8.

Answer: We agree with the referee’s suggestion and the corresponding Figures have been converted in linear trees

The English within the text needs significant improving, in particular improvements in grammar and word choice are required

Answer: The paper has been revised by an English mother tongue reader

I cannot find mention of sequence submission to a database such as NCBI. This should be done and the GenBank accession numbers included in the text

Answer: Because submission of the entire sequence data set to public databases would permit transmission networks to be identified and thus risk breaching patient confidentiality, following scientific and ethical reasons explained in other studies {Alizon et al (2010), Kouyos et al (2010), Esbjornsson et al. (2016)} we have submitted a random sample of 10% (representative of each subtype) to GenBank (with upcoming) under accession numbers. However, all other sequences are available on request. This sentence has been added in the manuscript in the Materials and Methods as “Sequence data availability” subsection.

More details on sequence amplification and sequencing should be included in the methods section.

Answer: More details on sequence amplification and sequencing have been included in the method section.

Minor points

Results of the univariate analysis are shown in table 3 but the multivariate analysis is given in the text. I found this confusing and think it would be better if the multivariate analysis was also included in the table

Answer: Multivariate analysis has been added in Table 3.

In table 2, only the subtypes with more samples have large clusters and it is likely that this is a sampling issue. This should be pointed out in the text.

Answer: We have added this point.

.

In table 1, it would be helpful to have a final column with the total or overall figures.

Answer: A final column has been added in the Table 1